# Molecular Characteristics of Enterovirus B83 Strain Isolated from a Patient with Acute Viral Myocarditis and Global Transmission Dynamics

**DOI:** 10.3390/v15061360

**Published:** 2023-06-12

**Authors:** Juan Song, Huanhuan Lu, Lin Ma, Shuangli Zhu, Dongmei Yan, Jun Han, Yong Zhang

**Affiliations:** 1State Key Laboratory of Infectious Disease Prevention and Control, Collaborative Innovation Center for Diagnosis and Treatment of Infectious Diseases, National Institute for Viral Disease Control and Prevention, Chinese Center for Disease Control and Prevention, Beijing 102206, China; songjuan@ivdc.chinacdc.cn; 2National Polio Laboratory and WHO WPRO Regional Polio Reference Laboratory, National Health Commission Key Laboratory of biosafety, National Institute for Viral Disease Control and Prevention, Chinese Center for Disease Control and Prevention, Beijing 102206, China; luhuanhuan0908@163.com (H.L.); zhusl@ivdc.chinacdc.cn (S.Z.); yandm@ivdc.chinacdc.cn (D.Y.); 3Yunnan Institute of Endemic Diseases Control and Prevention, No.5, Wenhua Road, Dali 671000, China; oir3538@dingtalk.com

**Keywords:** enterovirus B83, molecular characteristics, phylogenetics, recombination, transmission dynamics

## Abstract

This study determined the global genetic diversity and transmission dynamics of enterovirus B83 (EV-B83) and proposed future disease surveillance directions. Blood samples were collected from a patient with viral myocarditis, and viral isolation was performed. The complete genome sequence of the viral isolate was obtained using Sanger sequencing. A dataset of 15 sequences (from three continents) that had sufficient time signals for Bayesian phylogenetic analysis was set up, and the genetic diversity and transmission dynamics of global EV-B83 were analyzed using bioinformatics methods, including evolutionary dynamics, recombination event analysis, and phylogeographic analysis. Here, we report the complete genome sequence of an EV-B83 strain (S17/YN/CHN/2004) isolated from a patient with acute viral myocarditis in Yunnan Province, China. All 15 EV-B83 strains clustered together in a phylogenetic tree, confirming the classification of these isolates as a single EV type, and the predicted time for the most recent common ancestor appeared in 1998. Recombinant signals were detected in the 5’-untranslated region and 2A–3D coding regions of the S17 genome. The phylogeographic analysis revealed multiple intercontinental transmission routes of EV-B83. This study indicates that EV-B83 is globally distributed. Our findings add to the publicly available EV-B83 genomic sequence data and deepen our understanding of EV-B83 epidemiology.

## 1. Introduction

Human enteroviruses (EVs) are classified into four species: EV-A, EV-B, EV-C, and EV-D [1]. The species EV-B comprises the following 58 serotypes: coxsackievirus group B (CVB; serotypes 1–6); coxsackievirus group A (CVA; serotype 9); echovirus (serotypes 1–7, 9, 11–21, 24–27, and 29–33); EV-B69; and the recently identified EV serotypes designated as EV-B73–B75, EV-B77–B88, EV-B93, EV-B97–B98, EV-B100–B101, EV-B106–B107, and EV-B110–B114.

Among the EV serotypes, more than 85 are known human pathogens, although most human EV infections are asymptomatic or result in only mild diseases, such as the common cold; however, in some cases, EVs are the most common viral cause of serious illnesses, including acute myocarditis, acute flaccid paralysis, and aseptic meningitis. The viruses in the species EV-B, which are the most common viral causes of acute myocarditis, encephalitis, and aseptic meningitis [2,3,4], belong to the genus *Enteroviruses* in the family *Picornaviridae* and order *Picornavirales*. Picornaviruses are small non-enveloped EVs comprising 60 copies of each of the capsid proteins VP4, VP2, VP3, and VP1, which enclose a positive-sense single-stranded RNA genome. Viral RNA (approximately 7500 nucleotides) contains a long open reading frame (ORF) flanked by a 5′-untranslated region (UTR) and a 3′-UTR. The ORF encodes a single polyprotein translated from the viral genome, which gets cleaved into three polyprotein precursors: the structural polypeptide P1 (contains VP1-VP4 four proteins) and the non-structural polypeptides P2 (containing 2A–2C three proteins) and P3 (containing 3A–3D four proteins) [5]. Due to the lack of proofreading activity during genome replication, recombination commonly occurs in addition to mutation in the 5′-UTR and nonstructural polypeptide region.

Viral Myocarditis refers to localized or diffuse acute or chronic inflammatory lesions of myocardium caused by a viral infection, with different clinical manifestations. The clinical manifestations of patients with viral myocarditis depend on the extent and location of the lesion. Some infected persons may be asymptomatic. Common clinical symptoms include fever, body aches, sore throat, fatigue, nausea, palpitations, chest tightness, chest pain or precordial pain, etc. Most patients recover after appropriate treatment, and very few patients die in the acute phase due to serious arrhythmia, acute heart failure, and cardiogenic shock [6]. Multiple viruses can cause myocarditis, with viral infections causing intestinal and upper-respiratory-tract infections being the most common. Coxsackievirus B, ECHO virus, and poliovirus are common myocarditis-causing viruses, among which coxsackie virus B is the most important virus. Other pathogens include human adenovirus, influenza virus, parainfluenza virus, measles virus, mumps virus, Japanese encephalitis virus, herpes simplex virus, cytomegalovirus, etc. [7].

Newly identified EVs refer to a new type of EV that has not been discovered in the past but has been discovered recently. With the continuous advancement of technology and the establishment of EV surveillance networks in various countries worldwide, an increasing number of new types of EVs have been discovered and identified, known as “novel EVs”. Scientists continue to identify new types of EVs in different regions of the world and conduct research to characterize the genetic composition of newly identified EVs, including identifying genomic sequences and analyzing their genetic diversity. The study of these newly identified EVs helps to understand the evolution and transmission patterns of the virus and to develop more effective prevention and control strategies.

In this study, we report the nucleotide sequence of the complete genome of an EV-B83 that was recovered from a 28-year-old adult with acute viral myocarditis in 2004. EV-B83 is one of the newly identified enteroviruses, and its prototype strain (strain USA/CA76-10392) was confirmed and reported in 2007 [8]. So, in 2004, when we isolated the virus, we could not type the virus by using the enterovirus neutralization test and nucleotide sequence analysis, and people did not recognize the virus, thus hindering us from further studying the virus. After the identification of the EV-B83 prototype strain in 2007, some other EV-B83s were successively discovered and reported. As of May 31, 2023, there were 27 nucleotide sequences reported on GenBank which were identified in countries such as the USA [8], France [9], India [10,11,12,13], Bangladesh [14], Cambodia [15], and China [16,17,18]; however, only one full-length genome sequence (the prototype strain) was reported. This provides us with the opportunity to conduct in-depth research on its molecular epidemiology, which is still insufficient. In this study, the virus was identified via a comparison with EV prototype strains and partially characterized EV isolates. This study increases the publicly available EV-B83 nucleotide sequence and furthers our understanding of EV-B83’s molecular epidemiology.

## 2. Materials and Methods

### 2.1. Sample Collection, Viral Isolation, and Primary Identification

The patient was a 28-year-old adult with acute viral myocarditis in Yunnan Province in 2004. Blood samples were collected on the day of outpatient visit, and the serum was precipitated at 37 °C for 2 h. The serum samples were detected for enterovirus (including Coxsackievirus A, Coxsackievirus B, ECHO virus, and poliovirus), paraechovirus, human adenovirus, influenza virus, parainfluenza virus, measles virus, mumps virus, rubella virus, herpes simplex virus, EB virus, and cytomegalovirus by reverse-transcript polymerase chain reaction (RT-PCR) or PCR, and the results were all negative, except for enterovirus. Subsequently, a 200 μL serum sample was collected and inoculated onto human rhabdomyosarcoma (RD) and human laryngeal carcinoma (HEp-2) cell lines. EV-like cytopathic effects were observed in both RD and HEp-2 cell lines. The viral isolate strain S17/YN/CHN/2004, hereafter referred to as strain S17, was not neutralized by intersecting EV antisera pools (National Institute of Public Health and the Environment (RIVM), Bilthoven, The Netherlands), according to standard procedures [19]. 

Informed consent was obtained from the patient. The Ethics Review Committee of the National Institute for Viral Disease Control and Prevention (IVDC) of the Chinese Center for Disease Control and Prevention approved this study and confirmed that all methods were performed as per the standard guidelines.

### 2.2. Viral RNA Extraction and Reverse Transcription

Viral RNA was extracted from viral isolates, using a QIAamp Viral RNA Mini Kit (Qiagen, Valencia, CA, USA), and stored at −80 °C for further use. Furthermore, 1 μL (200 U) SuperScript II ribonuclease H-reverse transcriptase (Invitrogen, USA) was used to produce single-stranded cDNA from 5 μL of purified viral RNA. cDNA syntheses were primed with 7500A and 011 (Table 1) and performed at 42 °C for 2 h, followed by 60 °C for 15 min to inactivate the enzyme. Finally, RNA in an RNA:DNA hybrid was specifically degraded with 1 μL ribonuclease H (Promega, USA) at 37 °C for 30 min.

### 2.3. Full-Length Genome Amplification

Two long-distance polymerase chain reaction (PCR) amplifications were performed using a TaqPlus Precision PCR system (Stratagene, CA, USA), which comprised a blend of Stratagene-cloned Pfu DNA polymerase (proof reading) and Taq2000 DNA polymerase (non-proof reading). Reactions contained 5 μL of cDNA (see above), 0.1 mM of each deoxynucleoside triphosphate, 10 μL of TaqPlus buffer, 1.0 ng/μL of a forward (0001S48 or 008) and reverse (011 or 7500A) primer (Table 1), and 5 units TaqPlus enzyme in a 100 μL reaction. Amplification was performed for 30 cycles at temperature levels of 94 °C (30 s), 60 °C (30 s), and 72 °C (6 min), followed by another two temperature levels of 94 °C (1 min) and 72 °C (20 min). Primer pairs 008/7500A and 0010S48/011 (Table 1) were used for long-distance PCR. The expected amplicons were 4.99 kb and 3.40 kb, respectively.

### 2.4. Nucleotide Sequencing

The two long-distance PCR products were purified using the QIAquick Gel Extraction Kit (Qiagen, Germany). Cycle sequencing reactions were performed using version 3.1 of the BigDye terminator chemistry (Applied Biosystems), using the primers listed in Table 1. Sequencing was performed in both directions, using an ABI PRISM 3130 Genetic Analyser (Applied Biosystems), and every nucleotide position was sequenced at least once from each strand. The 5′ segment sequences were determined using the 5′ rapid amplification of cDNA ends core set (Takara Biomedicals) according to the manufacturer’s instructions.

### 2.5. EV-B83 Dataset Construction

Nine full-length or approximately full-length EV-B83 genomic sequences were retrieved from the GenBank database (with a length limit of 6000–7600 nt as of 30 April 2023). In addition to S17/YN/CHN/2004 (hereafter referred to as S17) isolated in this study, the number of whole-genome sequences of EV-B83 reached 10. Additionally, a search was conducted for the *VP1* sequence of EV-B83 in the GenBank database (as of 30 April 2023), removing sequences with many missing bases and low-quality sequences, resulting in 24 *VP1* sequences of EV-B83. A limited number of EV-B83 strains have been isolated from six countries, namely the USA [8], France [9], India [10,11,12,13], Bangladesh [14], Cambodia [15], and China [16,17,18], since 1999. Muscle software (version 3.8.31_i86linux32) was used for alignment, and RAxML software (version 8.2.12) was used to construct a maximum likelihood (ML) tree of *P1*, *P2*, and *P3* sequences [23,24]

### 2.6. Bayesian Temporal Dynamics Analysis

TempEst (version 1.5) was used to detect whether the sequences contained sufficient time signals [25]. Bayesian phylogenetic analyses can only be performed using sequences with sufficient time. Additionally, ModelGenerator 0.85 was used to obtain the optimal nucleotide substitution model for the dataset [26]. Bayesian phylogenetic analysis was performed using BEAST (version 1.8.4) [27]. Tracer software (version 1.7.1) was used to assess whether the parameters converged, and a valid sample size >200 indicated parameter convergence [28]. The Bayesian maximum class creditability (MCC) tree was ultimately generated using the TreeAnnotator software (version 1.8.4), and the top 10% of the sampled trees were removed with the burn-in option. Furthermore, the MCC tree was visualized using the FigTree software (version 1.4). The Bayesian Skyline Plot (BSP) method was used for the dynamic inference of group history.

### 2.7. Recombination Analysis

Phylogenetic relationships between EV-B83 strain S17 and other EV-B prototype strains were generated from nucleotide sequence alignment, using the ML algorithm of MEGA software (version 11.0) [29]. Using the Basic Local Alignment Search Tool (BLAST; https://blast.ncbi.nlm.nih.gov/Blast.cgi, accessed on 1 May 2023) for sequence alignment retrieval, sequences in the nucleotide database with over 85% similarity to the *P3* coding region of S17 were obtained to analyze potential recombination events. The Recombination Detection Program (RDP4, version 4.46) was used for preliminary recombination analysis [30], using the following seven methods: RDP, GENECONV, Chimaera, MaxChi, Bootscan, SiScan, and 3Seq for recombination detection. Recombination events supported by four or more methods were considered credible. Based on the results of the RDP4, SimPlot (version 3.5.1) (200-nt window moving in 20-nt steps) was used to determine the recombination events that occurred and the recombination skeleton [31].

### 2.8. Bayesian Phylogeographic Analysis

The Bayesian tip association significance testing (BaTS) program (version 2.0) is commonly used to assess the strength of the geographic clustering of data [32]. Typically, each sequence in a dataset is assigned a character state according to its isolated region. The sample trees generated by BEAST software were processed using BaTS software, and the following three indicators were obtained to evaluate the strength of clustering: association index, parsimony score, and maximum monophyletic clade. A *p*-value < 0.05 was considered a significant association. 

Reconstructing spatial transmission patterns is an important auxiliary method for tracing the origin of EV-B83. We performed a phylogeography analysis by using the BEAST software, selected the asymmetric substitution model, and used the Bayesian Stochastic Search Variable Selection method to infer social networks [33]. The migration pathway, posterior probability (PP), and Bayes factor (BF) between the different regions were generated using SpreadD3 (version 0.9.7.1) [34]. Notably, the supported migration paths should meet the criteria BF >3 and PP >0.50 [35].

### 2.9. Nucleotide Sequence Accession Number

The nucleotide sequence of the complete genome of the Chinese EV-B83 strain S17/YN/CHN/2004 was deposited in GenBank (Accession No. OQ990312).

## 3. Results

### 3.1. Whole-Genome Sequences Analysis of Strain S17

The genome length of strain S17 was 7396 nt, encoding a polypeptide of 2184 amino acids. The coding sequences were flanked by a non-coding 5′-UTR of 742 nt and a non-coding 3′-UTR of 100 nt. A poly(A)tail composed of a long sequence of adenine nucleotides was added at the 3’ end of the genome. The analysis of the complete genome sequence of strain S17 showed that its genome was collinear with that of the EV-B83 prototype strain (USA-CA76-10392), except for a deletion at 110 nt, two insertions at 92 and 152 nt in the 5-UTR region, and one insertion at 7303 nt in the 3-UTR region. The overall base composition of the S17 genome was 28.37% A, 24.12% G, 23.69% C, and 23.82% U. Furthermore, polypeptide cleavage sites were estimated using the complete genome sequence of the EV-B83 prototype strain. Table 2 shows the nucleotide and amino acid sequence identities among strain S17, the prototype strain, and other EV-B strains.

### 3.2. The Evolutionary Dynamics of EV-B83

The nucleotide sequences available for distinct EV-B83 strains differed considerably. The sequence of the *VP1* coding region of strain S17 displayed 77.2–92.4% nucleotide and 86.27–96.83% amino acid identities with previously described EV-B83 strains. Through screening, we found that 15 sequences had sufficient time signals for a Bayesian phylogenetic analysis, including one American strain, five Indian strains, and nine Chinese strains (one, two, and six from Guangdong Province, Yunnan Province, and the Tibet Autonomous Region, respectively). All EV-B83 strains clustered together in a phylogenetic tree, confirming the classification of these isolates as a single EV type. Fifteen EV-B83 sequences predicted the time of the most recent common ancestor to be 1998, with an evolution rate of 5.611 × 10^−2^ substitutions/site/year (95% highest posterior density: 0.0413–0.0703). The earliest occurrence of MCC trees was in a Chinese Tibetan strain isolated in 1999. After 2000, EV-B83 began to separate from other regions in China, the United States, and India, and the diversity of this time series changed. The BSP also shows that the population began to expand during this period. After 2004, the detection rate of the sequence decreased, and the diversity decreased. The Indian strain was primarily detected, which is consistent with the inference from the BSP. The population size began to decline after a stable period (Figure 1).

### 3.3. Recombination Analysis of S17

The phylogenetic tree analysis suggested that EV-B83 strains were only monophyletic in the capsid-coding (*P1*) region (Figure 2a), indicating recombination between EV-B83 and other EV-B types. Strain S17 was grouped with other enteroviral serotypes in EV-B in the nonstructural protein *P2* and *P3* coding regions (Figure 2b,c), indicating that recombination events may have occurred.

The BLAST analysis of *P3* region nucleotide sequences revealed that the potential recombinant parents of strain S17 included 28 sequences from the following 12 EV serotypes: CVB1, CVB2, CVB3, CVB4, CVB6, E6, E7, E11, E20, E30, EV-B79, and EV-B85, all belonging to EV-B, suggesting that strain S17 may have frequent recombination with EV-B. In order to confirm the recombination events and identify its potential parents, the data pool was composed of the possible recombination sequence from the BLAST analysis and the prototype strain sequence of EV-B. An RDP4 analysis was conducted on the above data pool, and the results showed that the American strain of EV-B83 (GenBank number AY843301) and the Chinese Guangdong strains of EV-B83 (GenBank number MN597453) might be the skeleton of the S17 recombination event. The American strain of EV-B83 recombined with CVB4 (GenBank accession number MF422559) and E20 (GenBank accession number KF812551), while the Chinese Guangdong strains of EV-B83 recombined with E7 (GenBank accession number AY896765). The Chinese Guangdong strain (GenBank number MN597453) of EV-B83 is considered the backbone of the S17 recombination event based on the analysis of lineage geography. EV-B83, isolated from Yunnan, China, was obtained from the Guangdong strain (GenBank number MN597453). Using SimPlot analysis, it was confirmed that the Chinese Guangdong strain (GenBank number MN597453) and E7 (GenBank number AY896765) recombined in the 5’- UTR and 2A-3D coding regions, leading to the production of strain S17 (Figure 3).

### 3.4. EV-B83 Phylogeographic Analysis

The phylogeographic analysis revealed that EV-B83 was introduced from Tibet, China, to Guangdong, China, between 2000 and 2001 (with a Bayesian factor value of 24). Subsequently, EV-B83 was introduced in the United States (with a Bayesian factor value of 5) and Yunnan, China (with a Bayesian factor value of 4), in 2001–2004 and 2002–2004, respectively. EV-B83, which was introduced in Yunnan, China, was introduced in India between 2008 and 2009 (with a Bayesian factor value of six), leading to the spread of EV-B83 in India (Figure 4).

## 4. Discussion

Mutations and recombination are the two major mechanisms involved in the evolution of EVs [36,37]. They continuously accumulate changes in the genome, enabling the virus to rapidly adapt to changes in the host immune system and environment. Both mutations and recombinations can cause changes in the EV genome, thereby affecting the biological characteristics and pathogenicity of the virus. Furthermore, EV evolution occurs through the accumulation of genetic changes via mutations and recombination, leading to changes in the biological characteristics and pathogenicity of the virus. Enteroviral mutations mainly refer to mutations that occur in the viral genome during replication and may affect the biological characteristics and immunogenicity of the virus [38,39]. Enteroviral recombination refers to the exchange of gene sequences between different viral strains, which may produce new genome combinations [40,41]. For example, when two different enteroviral strains infect the same host, their genomes may recombine during replication, resulting in a new genome recombination. Even in the newly identified EVs, traces of genomic recombination could still be found, indicating that almost all existing EV genomes have undergone recombination.

Sequence divergence is greatest in the *VP1* coding region because of the evolution of a given EV serotype. Indeed, molecular typing methods based on sequence variations in this region correlate with classifications based on antigenic properties and have recently replaced the neutralization test as the gold standard for EV typing [42,43]. In contrast, non-capsid regions do not correlate with enteroviral serotypes because of frequent recombination within these regions, with recombination usually occurring among enteroviral serotypes within a species [44,45]. The Chinese EV-B83 strain (S17) from this study is no exception; it has recombined with other EV-B viruses, and as several other reports have demonstrated EV-B recombination, it is impossible to identify precise parental sequences for the studied genome. Based on the findings of recombination analysis, it is reasonable to speculate that the long-term evolution of these viruses originated from the same ancestor, which provides the spatial and temporal circumstances for recombination to occur.

The sequences of 27 EV-B83 strains were available up until recently in the GenBank database (until 31 May 2023). A limited number of EV-B83 strains were isolated from three continents during 1999–2012, namely North America (USA), Europe (France), and Asia (China, Cambodia, Bangladesh, and India), indicating that EV-B83 infection has a global distribution, particularly in Asia. The substantial divergence between *VP1* coding sequences and the existence of several recombination events between non-capsid sequences led to the formation of several clusters in the phylogenetic analysis, suggesting the circulation of distinct EV-B83 lineages in different countries. 

Viral myocarditis is an inflammatory heart disease which mainly affects the myocardium. Enterovirus is one of the main pathogens that cause viral myocarditis by directly infecting myocardial cells, triggering inflammatory reaction, and leading to myocarditis [7,46]. Not all enterovirus infections can lead to viral myocarditis, but some enterovirus subtypes, especially coxsackievirus group B, especially CVB3, have a high risk of myocarditis [47,48]. Other enterovirus, such as echovirus and coxsackievirus group A, have also been reported to cause viral myocarditis [49,50]. Both CVB3 and EV-B83 belong to EV-B and have the potential to cause viral myocarditis. However, there is very little research on EV-B83, and there is an urgent need to strengthen the surveillance and research of this virus.

EV-B83 has rarely been reported; its global epidemiological and transmission characteristics are largely unknown, and its pathogenicity has been significantly underestimated. During the investigation of a patient with acute viral myocarditis in Yunnan, China, strain S17 was isolated from a blood sample. However, China has not yet established an effective EV etiology or symptom surveillance system for acute viral myocarditis. Therefore, there is currently no information about EV-B83 or other new EVs in the epidemic. Once this is in place, it will be possible to monitor the transmission trends of EV-B83 or other new EVs through large-scale molecular epidemiological research.

Although our study was limited by the lack of EV-B83 pathogenic surveillance and relatively few EV-B83 full-length genome sequences obtained, this work provides a foundation for understanding the epidemiological characteristics, phylogenetic features, and Bayesian phylodynamics of EV-B83. This study increases the number of publicly available EV-B83 genomic sequences and advances our understanding of EV-B83 molecular epidemiology.

## Figures and Tables

**Figure 1 viruses-15-01360-f001:**
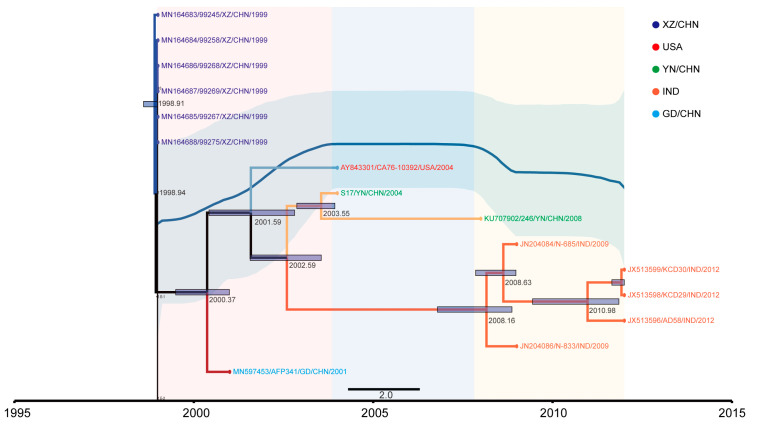
Temporal dynamics analysis of EV-B83 based on *VP1* sequence. Maximum clade credibility tree based on the *VP1* region of 15 EV-B83 sequences. The sequence branches are colored according to different locations.

**Figure 2 viruses-15-01360-f002:**
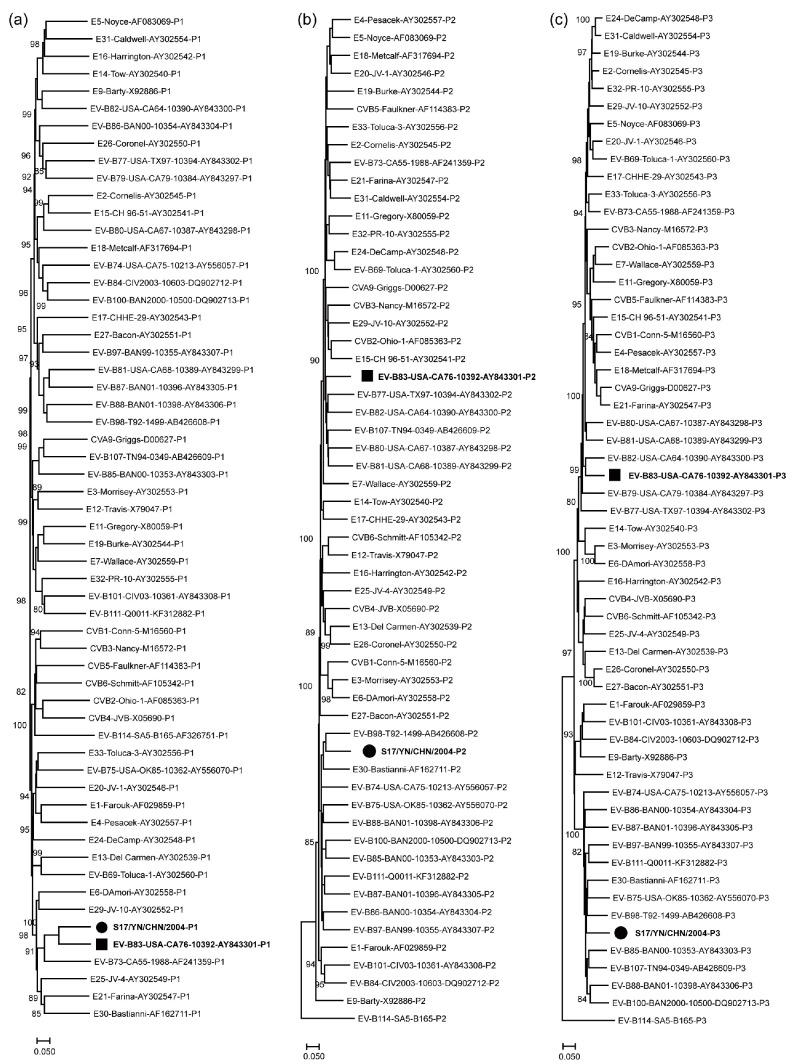
Phylogenetic relationships of EV-B83 strain S17 and other EV-B prototype strains. Numbers at nodes indicate bootstrap support for that node (per cent of 1000 bootstrap pseudo-replicates). The scale bars represent the genetic distance, and all panels have the same scale. (**a**) *P1* coding sequences. (**b**) *P2* coding sequences. (**c**) *P3* coding sequences. EV-B, enterovirus B83.

**Figure 3 viruses-15-01360-f003:**
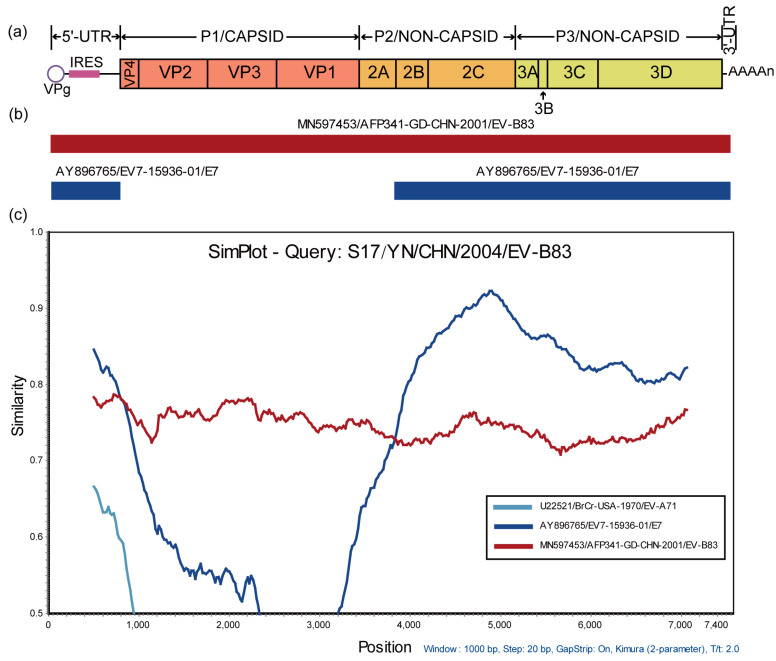
Recombination analysis of EV-B83 strain S17. (**a**) Schematic of the genome structure of enteroviruses. (**b**) Recombination analysis results of EV-B83 strain S17, using RDP4. (**c**) Recombination analysis results of EV-B83 strain S17, using Simplot. Bootstrap values above 70% were considered to reflect the presence of recombination events. RDP4, Recombination Detection Program.

**Figure 4 viruses-15-01360-f004:**
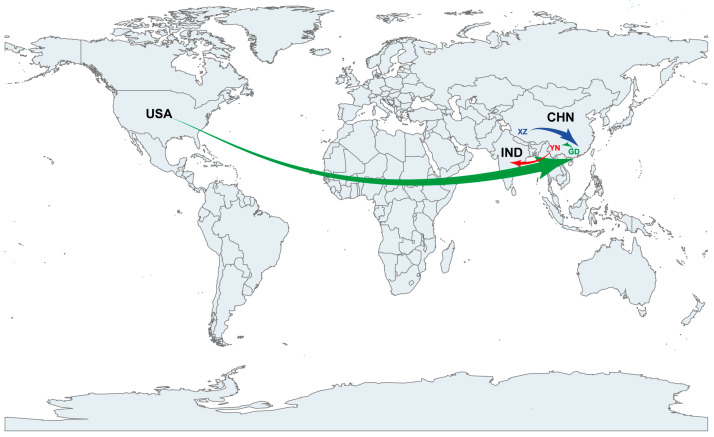
Phylogeographic analysis of EV-B83. Spatial diffusion pathways of EV-B83 in different countries. Only migration pathways with BF value > 3 and indicator > 0.50 are shown. BF, Bayes factor.

**Table 1 viruses-15-01360-t001:** PCR and sequencing primers.

Primer	Nucleotide Position (nt)	Primer Sequence	Orientation	Reference
0001S48 ^a^		GGGGACAAGTTTGTACAAAAAAGCAGGCTTTAAAACAGCTCTGGGGTT	Forward	[20]
EV/PCR-1	539–564	ACACGGACACCCAAAGTAGTCGGTCC	Reverse	[20]
EVP4	541–560	CTACTTTGGGTGTCCGTGTT	Forward	[21]
OL68-1	1178–1197	GGTAAYTTCCACCACCANCC	Reverse	[21]
S17-936Y	936–955	CTACCCGCATTGAACTCACC	Forward	This study
S17-1716Z	1697–1716	GAGGCGCAATCCATTGTACT	Reverse	This study
S17-1586Y	1586–1605	CCGGTCATTACAACTTCACC	Forward	This study
S17-2567Z	2548–2567	ATGGTGTCGCTGGGAACTAC	Reverse	This study
008	2411–2430	GCRTGCAATGAYTTCTCWGT	Forward	[22]
011	3389–3408	GCICCIGAYTGITGICCRAA	Reverse	[22]
S17-3167Y	3167–3186	GCCCACCACGTCTCTGTAAT	Forward	This study
S17-4245Z	4226–4245	TGAGGGTGCACTCTGCTCTA	Reverse	This study
S17-3981Y	3981–4000	GCGTTGGCTCAAACAAAAAG	Forward	This study
S17-4970Z	4951–4970	AACATTTCGGTCACGAGCAT	Reverse	This study
S17-4437Y	4437–4456	TGGAAAATCAGTGGCAACAA	Forward	This study
S17-5423Z	5404–5423	TATTCGGTTTTCACGGTGCT	Reverse	This study
S17-5277Y	5277–5296	GTTTGCAGGTTTCCAAGGAG	Forward	This study
S17-6348Z	6329–6348	TTTGTCCATGCACTCCTTCA	Reverse	This study
S17-6219Y	6219–6238	TGAAGGCCTAGAAGCACTGG	Forward	This study
S17-7353Z	7334–7353	GTTCGGTGAGTGTGGTAGGG	Reverse	This study
S17-6777Y	6777–6796	GTGTTCTGGGACCAGCATTT	Forward	This study
7500A ^a^		GGGGACCACTTTGTACAAGAAAGCTGGG(T)_24_	Reverse	[20]

^a^ The primer pairs 008/7500A and 0010S48/011 are suggested for long-distance PCR. The expected amplicons from these are 4.99 kb and 3.40 kb, respectively.

**Table 2 viruses-15-01360-t002:** Sequence properties of a Chinese EV-B83 strain (S17/YN/CHN/2004) and comparison of nucleotide and deduced amino acid sequences with EV-B83 prototype strain (CA76-10392) and other EV-B strains.

Genome Region	Position (nt)	Length (nt)	S17 vs. CA76-10392 (%)		S17 vs. Other EV-B Strains (%)		Note ^a^
			Nucleotide	Amino Acid	Nucleotide	Amino Acid	
5′-UTR	1–743	743	83.3	/	67.7–89.0	/	One Deletion, Two Insertions
*VP4*	744–950	207	77.7	91.3	70.5–77.2	73.9–94.2	
*VP2*	951–1727	777	79.9	98.0	66.2–71.7	73.6–84.2	
*VP3*	1728–2438	711	78.8	97.4	63.2–72.8	65.8–82.7	
*VP1*	2439–3290	852	74.9	94.0	52.7–65.8	54.6–69.3	
*2A*	3291–3740	450	79.7	95.3	74.8–82.8	90.0–96.0	
*2B*	3741–4037	297	81.4	96.9	74.4–87.2	91.9–96.9	
*2C*	4038–5024	987	80.8	96.9	79.0–86.6	96.3–98.4	
*3A*	5025–5291	267	77.5	96.6	74.1–87.2	92.1–100.0	
*3B*	5292–5357	66	75.7	90.9	68.1–86.3	81.8–95.4	
*3C*	5358–5906	549	79.5	95.6	75.5–86.1	92.8–98.9	
*3D*	5907–7292	1386	78.6	95.0	76.8–86.3	94.1–98.0	
3′-UTR	7296–7396	101	88.1	/	79.2–97.0	/	one insertion

^a^ Nucleotide deletions compared with EV-B83 prototype strain (CA76-10392). One deletion at nt 110, two insertions at nt 92 and nt 152 in the 5′-UTR region, and one insertion at nt 7303 in the 3′-UTR region were identified.

## Data Availability

The nucleotide sequence of the complete genome of the Chinese EV-B83 strain S17/YN/CHN/2004 was deposited in GenBank (Accession No. OQ990312).

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
