# Peer review of "Molecular Characteristics of Enterovirus B83 Strain Isolated from a Patient with Acute Viral Myocarditis and Global Transmission Dynamics"

_viruses, 2023, doi:10.3390/v15061360_

Round 1

Reviewer 1 Report

I have no major concerns about the manuscript other than the reason for completing the study needs additional context. The authors did not clearly explain why they are sequencing a virus sample that was isolated in 2004. I originally thought there was a mistake in the name of the virus since they listed the year as 2004.

While the findings are interesting and significant, I was confused why this research was being completed nearly 20 years after isolation of the sample. A paragraph explaining the rationale for sequencing this specific sample would help put the research into perspective.

Author Response

Reviewer 1

I have no major concerns about the manuscript other than the reason for completing the study needs additional context. The authors did not clearly explain why they are sequencing a virus sample that was isolated in 2004. I originally thought there was a mistake in the name of the virus since they listed the year as 2004.

While the findings are interesting and significant, I was confused why this research was being completed nearly 20 years after isolation of the sample. A paragraph explaining the rationale for sequencing this specific sample would help put the research into perspective.

------Response to reviewer 1 comment 1: It has been clarified in the manuscript. In this study, we report the nucleotide sequence of the complete genome of an EV-B83, which was recovered from a 28-year-old adult with acute viral myocarditis in 2004. EV-B83 is a newly identified enterovirus, and its prototype strain (strain USA/CA76-10392) was confirmed and reported in 2007. So in 2004, when we isolated the virus, we could not type the virus by using the Enterovirus neutralization test and nucleotide sequence analysis, and people did not recognize the virus, which hindered us from further studying the virus. After the identification of the EV-B83 prototype strain, some other EV-B83 were successively discovered and reported. As of May 31, 2023, there were 27 nucleotide sequences reported on GenBank, which were identified in countries such as USA, France, India, Bangladesh, Cambodia, and China, however, only one full-length genome sequence (the prototype strain) was reported. This provides us with the opportunity to conduct in-depth research on its molecular epidemiology, which is still insufficient. In this study, the virus was identified by comparison with EV prototype strains and partially characterised EV isolates. This study increases publicly available EV-B83 nucleotide sequence and further our understanding EV-B83 molecular epidemiology. (Page 2, lines 79-94).

Reviewer 2 Report

The manuscript of Song et al. describes the genetic analysis of an enterovirus B83 isolated from a patient with acute myocarditis. Additionally, the writers have, analyzed the transmission according to the complete genomic sequence of a few EV-B83 isolates. This is only the third publication adding to the two found in PubMed on the genetic analysis of an EV-B83. Furthermore, this virus was isolated from a myocarditis patient, which makes this publication not only novel, but also important for the health of humans.

This manuscript is mostly well written. However, there are minor points, which need correction or further description. The detailed comments are below:

Materials and methods:

Details of the patient and EV diagnostics, rows 67-74: there were probably other viral or bacterial diagnostics done? What microbial diagnostics was done and were there other findings? Was and EV RT-PCR done from either the patient serum sample or growth media, since no neutralization with EV type specific antisera was detected. Was EV sequencing PCR done just because of the EV-like CPE? I would like to see these added, if done.

The writers claim that the blood sample was collected at the day of disease onset. Was this really so, did the patient come on the same day as symptoms appeared?

Row 75: The sentence “Informed consent was obtained from all patients”. As there was only one patient, why not say “Informed consent was obtained from the patient”?

Row 166: The sequence cannot yet be found in Gene Bank.

Page 6 (or 1/13 as in the manuscript for a second time):

Table 2 and text after it, section 3.2. second sentence (no line numbers anymore). There are numbers for VP1 sequence identities, these are different in the text (77.23%-92.37% / 86.27%-96.83%) and table (52.7-65.8% / 54,6-69.3%). Also, why not use same accuracy (decimal places after shown). Please correct which are the correct numbers.

Page 8, 9, 10. after Figures 2, 3, 4: there are “sentences” out of place? What are these, do they belong in these places. Page 8 “EV-B, enterovirus B83”, page 9 “RDP4, Recombination Detection Program”, page 10 “BF, Bayes factor”

Page 8: 3rd to 5th row from beginning of the paragraph: “By combining the prototype strain sequence of EV-B EV, the scope of the recombination event was further narrowed using the RDP4 software.” What does here mean the underlined part? What was compared to what? Please revise to make clear.

Text of figures and tables: Looks like they are aligned with other text not with tables or figures.

Author Response

Reviewer 2

The manuscript of Song et al. describes the genetic analysis of an enterovirus B83 isolated from a patient with acute myocarditis. Additionally, the writers have, analyzed the transmission according to the complete genomic sequence of a few EV-B83 isolates. This is only the third publication adding to the two found in PubMed on the genetic analysis of an EV-B83. Furthermore, this virus was isolated from a myocarditis patient, which makes this publication not only novel, but also important for the health of humans.

This manuscript is mostly well written. However, there are minor points, which need correction or further description. The detailed comments are below:

Materials and methods:

Details of the patient and EV diagnostics, rows 67-74: there were probably other viral or bacterial diagnostics done? What microbial diagnostics was done and were there other findings? Was and EV RT-PCR done from either the patient serum sample or growth media, since no neutralization with EV type specific antisera was detected. Was EV sequencing PCR done just because of the EV-like CPE? I would like to see these added, if done.

------Response to reviewer 2 comment 1: It has been clarified in the manuscript. The serum samples were detected for enterovirus (including Coxsackievirus A, Coxsackievirus B, ECHO virus and poliovirus), paraechovirus, human adenovirus, influenza virus, parainfluenza virus, measles virus, mumps virus, rubella virus, herpes simplex virus, EB virus and cytomegalovirus by reverse-transcript polymerase chain reaction (RT-PCR) or PCR, and the results were all negative except for enterovirus. (Page 3, lines 99-103).

The writers claim that the blood sample was collected at the day of disease onset. Was this really so, did the patient come on the same day as symptoms appeared?

------Response to reviewer 2 comment 2: It has been clarified in the manuscript. Blood samples were collected on the day of outpatient visit. (Page 3, line 98).

Row 75: The sentence “Informed consent was obtained from all patients”. As there was only one patient, why not say “Informed consent was obtained from the patient”?

------Response to reviewer 2 comment 3: Revised as suggested. Informed consent was obtained from the patient. (Page 3, line 110).

Row 166: The sequence cannot yet be found in Gene Bank.

------Response to reviewer 2 comment 4: We have asked GenBank to release the sequence, with a release date of June 11th.

Page 6 (or 1/13 as in the manuscript for a second time):

------Response to reviewer 2 comment 5: The page numbers have been adjusted.

Table 2 and text after it, section 3.2. second sentence (no line numbers anymore). There are numbers for VP1 sequence identities, these are different in the text (77.23%-92.37% / 86.27%-96.83%) and table (52.7-65.8% / 54,6-69.3%). Also, why not use same accuracy (decimal places after shown). Please correct which are the correct numbers.

------Response to reviewer 2 comment 6: It has been clarified in the manuscript. The numbers of VP1 sequence similarity in the text are the results of comparing S17 with the previously described EV-B83 strains, and the numbers of VP1 sequence similarity in the table are the results of comparing S17 with other EV-B strains, and the same accuracy (keep one place after Decimal separator) were used in the manuscript.

Page 8, 9, 10. after Figures 2, 3, 4: there are “sentences” out of place? What are these, do they belong in these places. Page 8 “EV-B, enterovirus B83”, page 9 “RDP4, Recombination Detection Program”, page 10 “BF, Bayes factor”

------Response to reviewer 2 comment 7: It has been clarified in the manuscript. Page 8 “EV-B, enterovirus B83”, page 9 “RDP4, Recombination Detection Program”, page 10 “BF, Bayes factor” are the full names of abbreviations in the figures.

Page 8: 3rd to 5th row from beginning of the paragraph: “By combining the prototype strain sequence of EV-B EV, the scope of the recombination event was further narrowed using the RDP4 software.” What does here mean the underlined part? What was compared to what? Please revise to make clear.

------Response to reviewer 2 comment 8: It has been clarified in the manuscript. BLAST analysis of P3 region nucleotide sequences revealed that the potential recombinant parents of strain S17 included 28 sequences from the following 12 EV serotypes: CVB1, CVB2, CVB3, CVB4, CVB6, E6, E7, E11, E20, E30, EV-B79, and EV-B85, all belonging to EV-B, suggesting that S17 may have frequent recombination with EV-B. In order to confirm the recombination events and identify its potential parents, the data pool was composed of the possible recombination sequence from the BLAST analysis and the prototype strain sequence of EV-B. RDP4 analysis was conducted on the above data pool, and the results showed that the American strain of EV-B83 (GenBank number AY843301) and the Chinese Guangdong strains of EV-B83 (GenBank number MN597453) might be the skeleton of the S17 recombination event. (Page 8, lines 254-163).

Text of figures and tables: Looks like they are aligned with other text not with tables or figures.

------Response to reviewer 2 comment 9: It has been clarified in the manuscript. Text of figures and tables are aligned with tables or figures.
